# Steroid Resistance Associated with High MIF and P-gp Serum Levels in SLE Patients

**DOI:** 10.3390/molecules27196741

**Published:** 2022-10-10

**Authors:** Alberto Beltrán-Ramírez, José Francisco Muñoz-Valle, Jorge I. Gamez-Nava, Ana Miriam Saldaña-Cruz, Laura Gonzalez-Lopez, Alejandro Padilla-Ortega, Francisco I. Arias-García, Gabriela Athziri Sánchez-Zuno, Cesar Arturo Nava-Valdivia, Juan Manuel Ponce-Guarneros, Jesús Jonathan García-Galindo, Edsaúl Emilio Perez-Guerrero

**Affiliations:** 1Doctorado en Farmacología, Centro Universitario de Ciencias de la Salud, Universidad de Guadalajara, Guadalajara 44340, Mexico; 2Instituto de Investigación en Ciencias Biomédicas, Centro Universitario de Ciencias de la Salud, Universidad de Guadalajara, Guadalajara 44340, Mexico; 3Departamento de Fisiología, Centro Universitario de Ciencias de la Salud, Universidad de Guadalajara, Guadalajara 44340, Mexico; 4Servicio de Hematología, Hospital Civil Fray Antonio Alcalde, Guadalajara 44280, Mexico; 5Departamento de Microbiología y Patología, Centro Universitario de Ciencias de la Salud, Universidad de Guadalajara, Guadalajara 44340, Mexico

**Keywords:** MIF, P-gp, systemic lupus erythematosus, steroid-resistance, interaction, CHAID

## Abstract

Approximately 30% of patients with systemic lupus erythematosus (SLE) present steroid resistance (SR). Macrophage migration inhibition factor (MIF) and P-glycoprotein (P-gp) could be related to SR. This work aims to evaluate the relationship between MIF and P-pg serum levels in SR in SLE. Methods: Case–control study including 188 SLE patients who were divided into two groups (90 in the steroid-resistant group and 98 in the steroid-sensitive (SS) group) and 35 healthy controls. MIF and P-gp serum levels were determined by ELISA. Multivariable logistic regression and chi-squared automatic interaction detection (CHAID) were used to explore risk factors for SR. Results: The steroid-resistant group presented higher MIF and P-gp serum levels in comparison with the SS (*p* < 0.001) and reference (*p* < 0.001) groups. MIF correlated positively with P-gp (rho = 0.41, *p* < 0.001). MIF (≥15.75 ng/mL) and P-gp (≥15.22 ng/mL) were a risk factor for SR (OR = 2.29, OR = 5.27). CHAID identified high P-gp as the main risk factor for SR and high MIF as the second risk factor in those patients with low P-gp. Conclusions: An association between MIF and P-gp serum levels was observed in SR. CHAID identified P-gp ≥ 15.22 ng/mL as the main risk factor for SR. More studies are needed to validate these results.

## 1. Introduction

Systemic lupus erythematosus (SLE) is a global, chronic, autoimmune disease that is potentially fatal due to multiple organ damage [1]. The aim of the treatment in SLE is to achieve remission of the disease and decrease the relapse rate [2]. Recently, new therapies have been approved for use in SLE patients; however, the prescription of glucocorticoids (GCS) is still fairly frequent [3]. In a retrospective cohort study, Kariburyo et al. observed that GCS and hydroxychloroquine are the most commonly prescribed medications for SLE in the USA [4]. Furthermore, clinical practice guidelines in Latin America recommend using GCS as standard treatment for SLE [3,5].

Even though GCS are frequently prescribed, their consumption is accompanied by significant side effects after long-term use, including osteoporosis, hyperglycemia, insulin resistance, diabetes, high blood pressure, and central obesity, among others [6,7]. In addition, around 30% of patients develop steroid resistance, which can be related to multiple factors, including signaling through membrane transporters and cytokines such as macrophage migration inhibition factor (MIF) [8,9].

P-glycoprotein (P-gp) is a membrane transporter belonging to the ABC superfamily that has been widely studied in cancer and rheumatic diseases [10,11]. In relation to SLE, its overexpression in lymphocytes has been associated with disease activity as well as treatment resistance, including steroid resistance [12,13,14,15]. Furthermore, our study group observed a relationship between soluble P-gp and the failure of disease-modifying antirheumatic drugs (DMARDs) and immunosuppressive agents in rheumatic diseases [16,17].

Another molecule of high relevance in the pathogenesis of autoimmune diseases which has been associated with resistance to glucocorticoids is the macrophage migration inhibition factor (MIF) [18,19]. Wang et al. reported high expression of MIF in peripheral blood mononuclear cells (PBMC) from patients with GCS resistance. Furthermore, the association between serum levels of MIF and SLE activity has also been widely reported [20,21].

To the best of our knowledge, no studies have evaluated the role of the interactions between MIF and P-gp in the development of resistance to steroid treatment in patients with SLE. Therefore, this work aims to evaluate the relationship between MIF and P-pg serum levels and their association with steroid resistance in SLE.

## 2. Results

### 2.1. Patients and Reference Group

Table 1 describes the clinical characteristics of the SLE patients and the reference (control) group that took part in this study. These groups were similar in age, sex, and other sociodemographic variables. The SLE patients had a median age of 40 years (18–75) and a median disease duration of 4 years (1–44) and were primarily women (96.28%). Regarding the disease activity, 90 (47.87%) SLE patients had active disease, defined as an SLE disease activity index (SLEDAI) score of ≥4. Moreover, all the SLE patients were under a steroid scheme, and 87.77% were taking at least one immunosuppressor (azathioprine, mycophenolate, cyclophosphamide, chloroquine, or methotrexate).

### 2.2. MIF and P-gp Serum Levels Comparison

Figure 1 compares MIF and P-gp serum levels between the reference, steroid-resistant (SR), and steroid-sensitive (SS) groups, as well as MIF and P-gp serum levels with different GCS doses. We observed differences between the abovementioned three groups *(p* < 0.001). In the post hoc analysis, we observed that the MIF serum levels were higher in the SR group (median = 17.49 ng/mL, range = 0.125–53.41 ng/mL) compared to the SS (median = 3.91 ng/mL, range = 0.34–10.34 ng/mL; *p* < 0.001) and the reference groups (median = 13.57 ng/mL, range 0.125–39.96 ng/mL; *p* < 0.001). Furthermore, the SS group had higher MIF serum levels than the reference group (*p* = 0.005) (Figure 1A).

P-gp serum levels differed between the SS, SR, and reference groups (*p* < 0.001). Multiple comparison analysis showed that the SR group had higher P-gp serum levels (median = 22.45 ng/mL, range = 0.13–87.76 ng/mL) compared to the SS group (median = 11.02 ng/mL, range = 0.21–83.23 ng/mL; *p* < 0.001) and the reference group (median = 8.7 ng/mL, range = 3.53–14.98 ng/mL; *p* < 0.001). However, P-gp serum levels were similar in the SS group and the reference group (*p* = 0.24). These comparisons are shown in Figure 1B.

No differences were observed between MIF serum levels and the different GCS dose groups (Figure 1C). In contrast, P-gp serum levels were different in patients with different GCS doses (*p* = 0.009) (Figure 1D). After the post hoc analysis, we observed that patients with doses ≥ 20 mg/24 h had higher P-gp serum levels (median = 24 ng/mL, range = 0.24–87.76 ng/mL) than those with GCS < 10 mg/24 (median = 9.49 ng/mL, range = 0.32–83.23 ng/mL; *p* = 0.25). A difference was also observed when comparing the subgroup of SLE patients with doses of GCS < 10 mg to those with doses ≥10 and <20 mg (median = 14.61 ng/mL, range = 0.13–87.11 ng/mL; *p* = 0.027).

### 2.3. Correlations between MIF, P-gp, and Clinical Variables in SLE Patients

The correlations between MIF and P-gp serum levels and the clinical variables and laboratory values are shown in Table 2. MIF serum levels showed a weak positive correlation with disease activity (rho = 0.161, *p* = 0.028) but not with other variables, including GCS doses and disease activity. On the other hand, a positive correlation was observed between GCS doses and P-gp serum levels (rho = 0.242, *p* = 0.001). Furthermore, increased P-gp serum levels were correlated with higher SLEDAI scores (rho = 0.245, *p* = 0.001). For P-gp, the rest of the correlations were not statistically significant. However, we observed that serum levels of MIF and P-gp showed a positive correlation with a rho value of 0.411 and a *p*-value < 0.001.

### 2.4. Results of Receiver Operator Characteristic

The receiver operator characteristic (ROC) results showed that the best cut-off for identifying SR patients according to MIF was 15.75 ng/mL. Levels higher than this cut-off were considered high MIF serum levels. The sensitivity and specificity for this cut-off were 63.7% and 64.3%, respectively. The ROC plot identified an area under the curve equal to 0.619. The ROC curve for P-gp serum levels identified that concentrations of 15.22 ng/mL were the best cut-off for classifying patients as SR or SS. The performance for the ROC curves showed a sensitivity and specificity of 64% and 66%, respectively, and an area under the curve of 0.67. Therefore, high serum P-gp levels were considered as those higher than 15.22 ng/mL.

### 2.5. Steroid-Resistant (SR) and Steroid-Sensitive (SS) Comparison

This study included 90 SLE patients in the SR group and 98 SLE patients in the SS group. Age and gender proportions were similar in the SR group compared to the SS group (*p* = 0.66 and *p* = 0.51, respectively). However, we observed higher tobacco consumption in the SR group than in the SS group (22.22% vs. 10.20%, *p* = 0.04). The frequency of comorbidities was higher in the SR group than in the SS group (44.4% vs. 26.5%, *p* < 0.001).

Laboratory values, such as C3, C4, and CRP, were similar in both groups (*p* = 0.39, *p* = 0.44, *p* = 0.71, respectively). However, erythrocyte sedimentation rate (ESR) (*p* = 0.004) and proteins in 24 h urine (*p* = 0.001) were higher in the SR group compared to the SS group. Anti-nuclear antibodies were positive in a higher proportion of patients in the SR group than in the SS group (52.20% vs. 36.70%, *p* = 0.04).

Both groups shared similarities in pharmacological management with GCS. The use of prednisone (*p* = 0.66) or deflazacort (*p* = 0.66) was similar in the SR and the SS group. The use of azathioprine was higher in the SS group compared to the SR group (47.96% vs. 31.11%, *p* = 0.02). However, the proportion of SLE patients with a mycophenolate prescription was higher in the SR group than in the SS group (37.78% vs. 22.45%, *p* = 0.02). The use of at least one immunosuppressive agent was similar in both groups (*p* = 0.26). The proportion of SLE patients with high MIF and P-gp serum levels was higher in the SR group compared to the SS group (MIF: 27.7% vs. 18.6%, *p* = 0.003; P-gp: 28:7% vs. 17.5%, *p* < 0.001). No other differences were found in the rest of the comparisons (Table 3).

### 2.6. Unconditional Logistic Regression Models

Table 4 shows the results of the multivariable unconditional logistic regression models. The included predictors were age, disease evolution, ESR, use of immunosuppressive agents, MIF levels, and P-gp serum levels. The dependent variable in both models was the presence of steroid resistance. Model A included serum levels of MIF and P-gp as continuous variables, while model B included MIF and P-gp as high or low levels defined by the ROC curve. Model A showed that MIF serum levels (OR = 1.11, 95% CI: 1.03–1.20), P-gp serum levels (OR = 1.11, 95% CI: 1.04–1.19), ESR (OR = 1.05, 95% CI: 1.01–1.09), and tobacco consumption (OR = 2.51, 95% CI: 1.12–5.92) were independent risk factors for steroid resistance. The rest of the variables evaluated in the model were not significant.

Model B showed that high serum levels of MIF (OR = 2.29, 95% CI: 1.20–7.62) and P-gp (OR = 5.17, 95% CI: 2.08–13.5) were significant risk factors for steroid resistance. Furthermore, tobacco consumption and having at least one comorbidity were risk factors for the SR group (OR = 3.44 and 2.91, respectively). The rest of the variables evaluated were not relevant for the model.

### 2.7. CHAID Decision Tree

Figure 2 shows the influencing interaction factors for steroid resistance. The CHAID algorithm found a three-level tree with seven nodes, out of which, four were terminal. The four major predictor variables included in the model were P-gp serum levels, MIF serum levels, presence of comorbidities, and the use of azathioprine. The root node had the P-gp serum levels as the main predictor for steroid resistance and split into two branches, those with serum levels of P-gp < 15.22 ng/mL and those with a level ≥ 15.22 ng/mL. In the second level of the tree, MIF serum levels and comorbidities were shown to be the best predictors for the different P-gp serum levels previously split in the first level. The terminal node three (*n* = 67) for patients with P-gp < 15.22 ng/mL and MIF < 15.75 ng/mL serum levels showed that an estimated 30% of SLE patients had steroid resistance, whereas more than 50% of patients with MIF levels ≥ 15.75 ng/mL presented steroid resistance in the terminal node four (*n* = 34). Presence of comorbidities was the second major predictor in the second branch of the first level, which was characterized by patients with ≥15.22 ng/mL of P-gp serum levels. Patients with P-gp serum levels ≥ 15.22 without comorbidities (node six, *n* = 62) had a similar percentage of steroid resistance compared to node four, in contrast to ≥85% of patients with steroid resistance in node seven (*n* = 25), who had higher P-gp serum levels and the presence of comorbidities.

## 3. Discussion

This study evaluated the possible relationship between MIF and P-gp serum levels and steroid resistance in SLE patients. We observed that the proportion of SLE patients with high MIF and P-gp serum levels was increased in the SR group compared to the SS group. This association was maintained after adjusting for confounding variables in the logistic regression model. Furthermore, the decision tree identified P-gp high serum levels as the primary conditioning risk factor for steroid resistance in SLE patients.

Even though steroid resistance can be the consequence of multiple factors [9,22,23], recent studies have found that MIF is a cytokine that contributes significantly to steroid resistance [18,20]. MIF confers resistance to steroids by suppression of the nuclear factor kB (NF-kB) inhibitor (IkB) and mitogen-activated protein kinase (MAPK) inactivator MAPK phosphatase-1; furthermore, it reverts the cytokine-induced suppression of cytoplasmatic phospholipase A2 provoked by GCS and activates the release of arachidonic acid, which results in an increase in inflammation due to the transcription of pro-inflammatory cytokines such as TNF and IL-6 [19,24]. Zhu et al. observed, in non-rheumatic disease, that non-responders to GCS had higher levels of MIF [25]. Furthermore, the evaluation of MIF serum levels and their association with GCS resistance in SLE has been reported. Wang et al. observed in a study with 62 SLE patients that patients with steroid resistance had higher MIF serum levels compared to patients with adequate response to steroids [20]. Our results were similar to these observations, reinforcing the hypothesis on the critical role of serum MIF in steroid resistance. However, further studies are needed to better understand the role of serum MIF in steroid resistance. Intriguingly, we did not find a correlation between the MIF serum levels and GCS doses. We were also unable to detect differences in serum concentrations of MIF in patients with different doses of GCS. This lack of association could be explained by the fact that exogenous GCS administration counter-regulates MIF levels [26]. Furthermore, it has been reported that MIF serum levels are lower in RA patients undergoing oral administration of steroids compared to patients not taking oral steroids [27].

Another molecule participating in steroid resistance is P-gp, a member of the ATP-binding cassette (ABC) transporters superfamily. P-gp is coded by the multidrug resistance 1 (MDR1) gene and functions as an efflux pump in different cells; therefore, it has been widely studied in patients with therapeutic resistance in several diseases [28,29]. In this study, we observed that high P-gp serum levels in SLE patients conferred a significant risk of steroid resistance. Although only a few studies have evaluated the serum form of P-gp [16,17], there is evidence of membrane expression that is consistent with the results presented here [12,14,15]. In a study carried out with SLE patients naïve to GCS, Kansal et al. observed that the membrane expression of P-gp in peripheral blood lymphocytes was higher in GCS non-responder patients compared to SLE patients with GCS response [15]. Tsujimura et al. observed increased P-gp expression in the PMBCs of SLE patients with high disease activity, regardless of GCS therapy [12]. Furthermore, in a cohort study by Zhang et al., the functionality and membrane expression of P-gp was enhanced in patients with persistent activity in contrast to those with adequate response [14].

We observed, in contrast to MIF serum levels, that serum levels of P-gp correlated positively with GCS doses. These results are in accordance with other reports [15,30] and could be explained by the induction of Y box binding protein 1, a transcription factor that upregulates the expression of P-gp mediated by GCS [31]. Furthermore, the induction of Y box binding protein 1 by GCS explains the lack of difference between the P-gp serum levels of SS SLE patients and those of the reference group. Thus, our results suggest that the serum form, as well as the membrane expression, are dependent on the GCS dose.

In this study, we were interested in the P-gp serum form described originally by Chu et al. [32]. Some studies have shown a direct relationship between soluble P-gp and its expression at the membrane level [32,33,34,35]. The P-gp soluble form has the same molecular weight (170 kDa) as P-gp expressed in the membrane; therefore, it has been proposed that P-gp is released into the extracellular space by a mechanism that does not involve proteolytic cuts [35]. Although the function of this serum form of P-gp is still rather unclear, our results could provide evidence that serum P-gp is related to steroid resistance in SLE. Our group previously related increased P-gp serum levels to disease activity as well as resistance to immunosuppressor agents and DMARDs in RA and SLE patients [16,17]; the findings of this study support this perspective by showing that serum P-gp could have an essential role in steroid resistance in rheumatic diseases such as SLE. Some studies reported that smoking can modify the expression of P-gp, but there is no evidence of this process with MIF [36,37]. In this study, we identified a major prevalence in smoking in the SR group, and this association remained after adjusting the model with confounding variables; however, when we analyzed the CHAID decision tree, this association did not have statistical significance. This is why we think that smoking has a relationship with P-gp and MIF but not with steroid resistance.

However, studies with different designs and approaches are needed to understand the actual role of this serum protein and its participation in pharmacological resistance.

Another remarkable result in this study was the positive correlation between P-gp serum levels and GCS doses. Patients with GCS doses ≥ 20 mg/day showed higher P-gp serum levels. These results are consistent with another study [12]. This positive association could be due to the fact that GCS are P-gp membrane inductors and, therefore, can develop an overexpression in the lymphocyte membrane [12]. This phenomenon added to a pro-inflammatory environment can lead to the efflux of different drugs, such as GCS and immunosuppressors, from the lymphocytes [38,39].

To the best of our knowledge, no studies have jointly evaluated MIF and P-gp serum levels in steroid resistance. One significant result is that, even in the presence of multiple confounding variables (tobacco use, age, etc.), higher concentrations of serum MIF and P-gp were risk factors for steroid resistance. Furthermore, in the CHAID decision tree, higher P-gp serum levels were the leading risk factor used to identify SLE patients with steroid resistance. However, for patients with P-gp serum levels lower than 15.22 ng/mL, the main risk factor was high MIF serum levels, which suggests that these molecules could be related. Another interesting result of our study, which, to our knowledge, has not been previously reported, is that patients with higher serum levels of P-gp also presented higher levels of MIF.

A possible explanation for the relationship found between the levels of MIF and P-gp could be the ability of MIF to induce the expression of other proinflammatory cytokines such as TNF-α and IL-6. This has been observed in PBMCs from SLE patients in vitro [40]. It has been reported that proinflammatory cytokines such as TNF-α can induce P-gp expression through the STAT3/NF-κB pathway [41]. Hence, the upstream role of MIF in the inflammatory cascade in inducing the expression of these proinflammatory cytokines could indirectly lead to an increase in the expression of P-gp in these patients. Both molecules in turn have a particular and well-discerned role in steroid resistance, which, in combination, reinforces our hypothesis that there is a possible relationship between P-gp and MIF in steroid resistance. Nevertheless, specifically designed studies to evaluate the molecular expression of MIF and P-gp are needed to establish if this statistical relationship is significant on a biological level. However, the CHAID analysis showed that patients with both P-gp serum levels higher than 15.22 ng/mL and comorbidities presented a high probability of steroid resistance, suggesting that there are other factors that can contribute to resistance to GCS in addition to these two molecules. Nonetheless, this study was not appropriately designed to identify these factors.

An important strength of this study was the use of two multivariable methods to explore influencing factors and their relationship with sensitivity or resistance to GCS in SLE patients for the first time. This study could be a turning point as we present the first evidence of a relationship between these two molecules and GCS resistance.

Our main results are based on MIF and P-gp serum levels; however, whether this statistical relationship also relates to molecular events inside the cell remains undescribed. Thus, these molecules may contribute to steroid resistance at a molecular level, but confirming this requires a different study approach. Another limitation observed is that the sample size may not have been enough to identify other variables associated with steroid resistance in the CHAID analysis, especially in the terminal nodes. The design of this study is another limitation because we were unable to calculate relative risk, an essential measure for identifying conferred risk of steroid resistance, or the variability of the serum levels of MIF and P-gp. Finally, we could not rule out disease activity as an influential factor in the higher levels of both molecules.

## 4. Materials and Methods

### 4.1. Study Design

This study was a case–control study. One hundred and eighty-eight patients classified according to the 1982 American College of Rheumatology criteria for SLE were included [42]. Patients were included if they met the following criteria: (a) regular use and stable dose of GCS for at least six months before the onset of this study, (b) ≥18 years old. SLE patients were excluded if they met any of the following criteria: (a) pregnancy, (b) another rheumatic or autoimmune disease, (c) change in the pharmacological regimen, including changes in doses or adding new drugs in the last six months, (d) presence of other diseases including chronic or acute infections, epilepsy, or cancer, (e) patients taking a drug acting as a P-gp inhibitor [31]. The reference control group consisted of 35 age-matched individuals with no infectious, rheumatic, chronic, or inflammatory disease. SLE patients and the reference group were recruited from the Centro Universitario de Ciencias de la Salud outpatient research department, University of Guadalajara.

### 4.2. Clinical Assessment

Trained researchers blinded to the MIF and P-gp serum levels performed a structured review of the epidemiological, clinical, and therapeutic characteristics of the patients involved in this study.

Systemic lupus erythematosus disease activity index (SLEDAI) score was used to assess the patient’s disease activity [43]. SLE patients were considered active if the SLEDAI score was ≥4. The disease activity was evaluated by a rheumatologist who was also blinded to the serum levels of both molecules.

Patients were subclassified into two groups according to their response to GCS therapy; the steroid-resistant (SR) group was composed of patients with SLEDAI index score ≥ 4 receiving a stable dose of ≥10 mg of prednisone, or its equivalent, for at least six months before the study onset; the steroid-sensitive (SS) group included the SLE patients with SLEDAI score ≤ 3 and stable doses of prednisone ≤ 10 mg, or its equivalent, for at least six months. Patients who did not meet the definitions described above were not included in this study. Furthermore, to compare MIF and P-gp serum levels at different GCS doses, we divided the patients into three groups based on their prednisone dose: <10 mg, ≥10 mg and <20 mg, and ≥20 mg.

### 4.3. Determination of MIF and P-gp Serum Levels

On the same day as their clinical evaluation, blood samples were collected from all SLE patients as well as the individuals included in the reference group. Samples were taken by trained staff in the early hours of the morning with 8 h of prior fasting. Samples were stored at −80 °C until processing. The investigators who performed the serum P-gp and MIF determinations were blinded to the clinical evaluations of the patients. All the samples were evaluated in duplicate following the manufacturer’s instructions. The MIF and P-gp serum levels were determined with commercial ELISA kits (Bio-Legend, USA, Cat. No. 438407-R09 and My BioSource, USA, Cat. No. MBS2703598, respectively). The MIF assay sensitivity was 17.4 pg/mL, and the P-gp assay sensitivity was 0.122 ng/mL.

### 4.4. Statistical Analysis

According to the distribution of the data, non-parametric statistics were used. For descriptive statistics, qualitative variables were expressed as frequencies and percentages; quantitative variables were expressed as medians and ranges. For comparison between qualitative and quantitative variables, chi-square and Mann–Whitney U tests were used, respectively. The Kruskal–Wallis test was performed for comparison between three groups, and the Bonferroni correction test was used for multiple comparisons in the post hoc analysis. For correlation analysis of quantitative variables, we used the Spearman correlation test.

ROC curves were used to determine high serum levels of MIF and P-gp. We selected the cut-offs according to performance in specificity and sensitivity to better identify steroid resistance.

An unconditional logistic regression was performed to evaluate the steroid resistance according to the serum levels of MIF and P-gp and other variables. The models included variables with a value under 0.2 in the univariate analysis or those with biological plausibility for steroid resistance. The final model was selected using a stepwise model, considering the values of the Akaike information criterion and Bayesian information criterion.

A CHAID decision tree was used to identify the SR group’s characteristics using possible predictor variables at different levels. Classification rules were selected as follows: (1) tree growth: the significance level of the growth branch segmentation α merge = 0.05, α split = 0.05; (2) the maximum depth of the tree was four levels; (3) the minimum sample number of the leaf node was 20; (4) the stopping rule was α = 0.05. If the node did not reach the sample size or the node was the leaf node, it was not split. The variables included in the CHAID tree were those with clinical plausibility for SLE patients: MIF and P-gp serum levels, age, disease evolution, tobacco use, comorbidities, SLICC, azathioprine or mycophenolate use, anti-dsDNA, and ANA. Age and disease evolution were included as factors with four levels according to the quantile distribution (25%, 50%, 75%).

Differences were considered significant at *p*-value ≤ 0.05. Statistical analyses were performed using R version 4.1.2 (R Core Team (2021). R: A language and environment for statistical computing. R Foundation for Statistical Computing, Vienna, Austria. URL https://www.R-project.org/, accessed on 13 January 2022).

## 5. Conclusions

In this study, the serum levels of MIF and P-gp were associated with steroid resistance in SLE patients, and this association remained even after adjusting for multiple confounding variables. The CHAID analysis identified that patients with higher P-gp serum levels have higher chances of steroid resistance, and high MIF serum levels significantly contribute to steroid resistance for those with lower levels of P-gp. A cohort study could reinforce these results by first obtaining the baseline serum levels of both molecules and then evaluating increases through follow-up investigations.

The results obtained in the present study are of great value for understanding the intra-individual variability in the response to steroid treatment in SLE, postulating that both molecules, MIF and P-gp, are possible therapeutic targets for reversing resistance to this treatment and improving the disease prognosis in these patients.

## Figures and Tables

**Figure 1 molecules-27-06741-f001:**
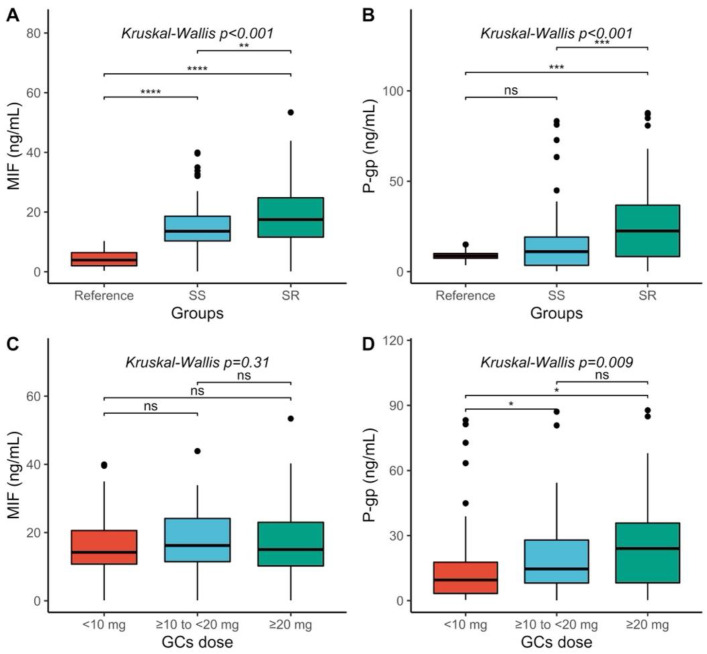
Comparison of MIF and P-gp serum levels; (**A**) comparison of the MIF serum levels among study groups; (**B**) comparison of P-gp serum levels among study groups; (**C**) comparison of MIF serum levels by GCS doses; (**D**) comparison of P-gp serum levels by GCS doses. ****: *p*-value ≤ 0.00001; ***: *p*-value ≤ 0.001; **: *p*-value ≤ 0.01; *: *p*-value ≤ 0.05; ns = not significant. SS, Steroid Sensitive; SR, Steroid Resistant.

**Figure 2 molecules-27-06741-f002:**
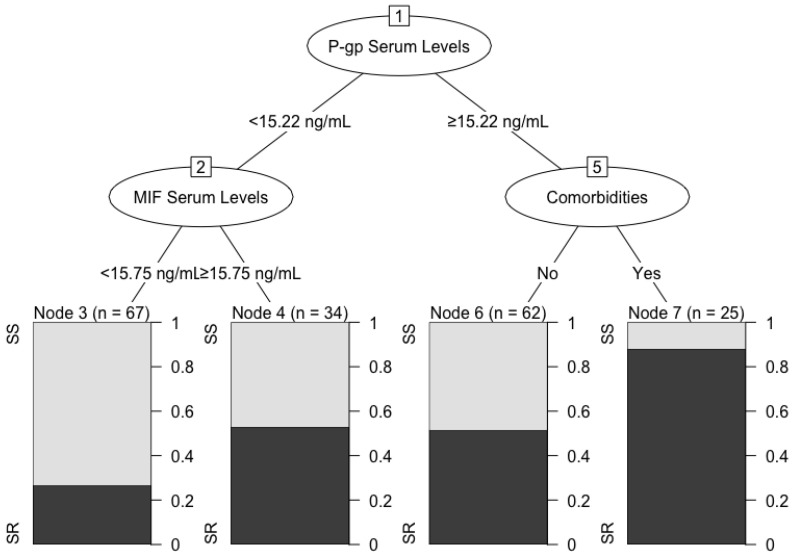
CHAID decision tree.

**Table 1 molecules-27-06741-t001:** Clinical characteristics of the SLE patients and the reference group in this study.

	SLE Patients (*n* = 188)	Reference Group (*n* = 35)	*p*-Value
Age (years)	40 (18–68)	49 (22–67)	<0.001
Women, *n* (%)	181 (96.28)	35 (100)	0.52
Alcohol use, *n* (%)	31 (16.49)	5 (14.28)	0.94
Tobacco use, *n* (%)	30 (15.96)	5 (14.28)	0.99
Comorbidities, *n* (%)	66 (35.11)	11 (31.42)	0.17
Disease evolution (years)	4 (1–28)	-	-
SLEDAI, active, *n* (%)	90 (47.87)	-	-
Glucocorticoids, *n* (%)	188 (100)	-	-
Prednisone, *n* (%)	166 (88.30)	-	-
Deflazacort, *n* (%)	22 (11.70)	-	-
Immunosuppressive agents, *n* (%)	165 (88.77)	-	-
Azathioprine, *n*, (%)	75 (39.89)	-	-
Mycophenolate, *n* (%)	56 (29.79)	-	-
Cyclophosphamide, *n* (%)	10 (5.32)	-	-
Chloroquine, *n* (%)	79 (42.02)	-	-
Methotrexate, *n* (%)	15 (7.98)	-	-

SLEDAI (systemic lupus erythematosus disease activity index) considered active if score ≥4; comorbidities include diabetes mellitus 2, high blood pressure, and osteoporosis.

**Table 2 molecules-27-06741-t002:** Correlations between MIF, P-gp, and clinical variables in SLE patients.

	MIF		P-gp	
	Rho	*p*-Value	Rho	*p*-Value
MIF, ng/mL	--	--	0.411	<0.001
P-gp, ng/mL	0.411	<0.001	--	--
Age, years	−0.036	0.627	0.013	0.855
Disease evolution, year	0.021	0.774	0.132	0.073
GCS doses	0.042	0.564	0.242	0.001
SLEDAI score	0.161	0.028	0.245	0.001
SLICC score	−0.116	0.113	0.001	0.987
C3 (mg/dL)	−0.038	0.661	0.124	0.145
C4 (mg/dL)	−0.062	0.468	0.027	0.753
ESR (mg/dL)	0.111	0.192	0.125	0.142
CRP (mm/hr)	−0.039	0.644	0.047	0.581
Protein in urine	0.118	0.166	0.099	0.245

MIF: macrophage migration inhibition factor; P-gp: P-glycoprotein; GCs: glucocorticoids; SLICC: Systemic Lupus International Collaborating Clinics; ESR: erythrocyte sedimentation rate; CRP: C-reactive protein.

**Table 3 molecules-27-06741-t003:** Steroid-resistant and steroid-sensitive group comparison.

	Steroid-Sensitive (*n* = 98)	Steroid-Resistant (*n* = 90)	*p*
Age (years)	42 (18–68)	38 (18–68)	0.66
Women, *n* (%)	93 (94.90)	88 (97.78)	0.51
Alcohol use, *n* (%)	14 (14.29)	17 (18.89)	0.51
Tobacco use, *n* (%)	10 (10.20)	20 (22.22)	0.04
Comorbidities, *n* (%)	26 (26.5)	40 (44.4)	<0.001
Disease evolution (years) ^a^	4 (1–20)	4 (1–28)	0.53
SLEDAI, active, *n* (%)	0 (0.00)	90 (100)	-
C3 (mg/dL) ^a^	106 (54–152)	104 (40–199)	0.39
C4 (mg/dL) ^a^	26.91 (8–72.69)	23.75 (8–275)	0.44
CRP (mg/dL) ^a^	7.67 (3.19–263.52)	8.9 (0.42–70.09)	0.71
ESR (mm/h) ^a^	23.59 (2–46)	29.38 (5–52.30)	0.004
Proteins in urine (mg/dL/24 h) ^a^	0.30 (0.04–10.77)	0.35 (0.01–10.77)	0.01
ANA, positive, *n* (%)	36 (36.70)	47 (52.20)	0.04
Anti-dsDNA, *n* (%)	18 (18.35)	24 (26.60)	0.23
Glucocorticoids, *n* (%)	98 (100)	90 (100)	1.00
Prednisone, *n* (%)	88 (89.90)	78 (86.67)	0.66
Deflazacort, *n* (%)	10 (10.20)	12 (13.33)	0.66
Glucocorticoid doses (mg/24 h) ^a^	7.2 (2.5–25)	17.5 (10–75)	<0.001
Immunosuppressive agents, n (%)	83 (84.69)	82 (91.11)	0.26
Azathioprine, *n,* (%)	47 (47.96)	28 (31.11)	0.02
Mycophenolate, *n* (%)	22 (22.45)	34 (37.78)	0.03
Cyclophosphamide, *n* (%)	2 (2.04)	8 (8.89)	0.05
Chloroquine, *n* (%)	42 (42.86)	37 (41.11)	0.92
Methotrexate, *n* (%)	8 (8.16)	7 (7.78)	1.00
High MIF serum levels, *n* (%)	35 (18.6)	52 (27.7)	0.003
High P-gp serum levels, *n* (%)	33 (17.5)	54 (28.7)	<0.001

^a^ = Quantitative variables expressed in medians and ranges; SLEDAI and SLEDAI-2KG considered active if score ≥ 4; ANA: antinuclear antibodies; Anti-dsDNA: anti-double-stranded DNA antibody; high MIF serum levels ≥ 15.75 ng/mL; high P-gp serum levels ≥ 15.22 ng/mL.

**Table 4 molecules-27-06741-t004:** Factors associated with steroid resistance in SLE.

	Model A		Model B	
Predictors	OR	95% CI	OR	95% CI
MIF serum levels	1.11	1.03–1.20	2.29	1.20–7.62
P-gp serum levels	1.11	1.04–1.19	5.17	2.08–13.5
Tobacco use	3.27	11.2–5.92.	3.44	1.40–9.10
Comorbidities	3.07	1.37–7.27	2.91	1.48–5.92
ESR	-	-	-	-
Age	-	-	-	-
Disease evolution	-	-	-	-
Immunosuppressive agents	-	-	-	-
CRP	-	-	-	-
Anti-dsDNA (positive)	-	-	-	-
ANA (positive)	-	-	-	-
SLICC ACR score	-	-	-	-

For model A: R^2^ = 0.261. For model B: R^2^ = 0.221. OR: odds ratio; CI: confidence interval. Comorbidities include diabetes mellitus 2, high blood pressure, and osteoporosis; ACR: American College of Rheumatology.

## Data Availability

The data presented in this study are not publicly available.

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
