# Peer review of "Steroid Resistance Associated with High MIF and P-gp Serum Levels in SLE Patients"

_molecules, 2022, doi:10.3390/molecules27196741_

Round 1
Reviewer 1 Report
1. Responses to therapies should be defined by SRI (the reduction of SLEDAI scores) but not SLEDAI or the dose of drug.
2. As authors mentioned in the introduction, P-gp is a cell membrane protein and plays important roles in multidrug resistance. The focus on serum P-gp levels in this study is a little bit weird.
3. It has been shown that P-gp is upregulated on lymphocytes from patients with active SLE and steroids suppress disease activity by inducing apoptosis. The application of high dose steroids might explain the elevation of P-gp in the serum.
Author Response
Guadalajara, Mexico, October 6th, 2022
We thank the reviewer 1 questions and suggestions regarding to our manuscript molecules-1943860, entitled: Steroid resistance associated with high MIF and P-gp serum levels in SLE patients.
We are submitting the new version of the manuscript above-mentioned. All changes were marked with track changes function in MS Word. All the recommendations and suggestions made by the reviewers were taken into account, and the questions were answered. In the following lines, we describe point by point the details of the revisions to the manuscript and our responses to the referees´ comments.
General modifications:
We submitted the manuscript to the MDPI english editing service, ID: english-51661. This is a new and improved version of the manuscript, with all the corrections made by your MDPI English editing services.
Reviewer 1 questions, observations, and suggestions:
- Responses to therapies should be defined by SRI (the reduction of SLEDAI scores) but not SLEDAI or the dose of drug.
Thank you for your comment.
The Systemic Lupus Erythematosus Responder Index (SRI) is an index that combines the application of other indices to assess disease activity. The SRI has been used mainly for the evaluation of efficacy in clinical trials for the evaluation of biological agents in SLE (Lupus Sci Med. 2018 Nov 26;5(1):e000288; Arthritis Rheum. 2009 Sep 15;61(9 ):1143-51). Although it has shown promising results in the context of clinical trials, the application of the SRI in daily clinical practice is not complicated due to the following aspects: 1) the cut-off point may be too strict, which makes it challenging to identify slight clinical improvements; 2) the time required to apply it is too long. (Autoimmun Rev. 2012 Mar;11(5):326-9; Arthritis Care Res (Hoboken). 2020 Oct;72 Suppl 10:27-46). This study used the SLEDAI, an index validated and widely used in the clinical context (Reumatol Clin. 2014 Sep-Oct;10(5):309-20). Although all the indices for evaluating the disease in SLE have advantages and disadvantages, for this study, the SLEDAI is an excellent tool to identify patients with resistance to corticosteroids.
On the other hand, this study was conceived as a case-control study with the aim of trying to identify the most significant number of patients without an adequate response to treatment with glucocorticoids. For this, it was proposed to consider as failure (case) those patients who maintained a high activity (SLEDAI≥4) despite having constant doses of steroids. Other studies have already used this strategy to identify patients with glucocorticoid failure (Arthritis Res Ther. 2012 May 2;14(3): R103), so we consider this an excellent strategy to compare our results with those reported.
We consider the observation about the use of SRI very importantly
- As authors mentioned in the introduction, P-gp is a cell membrane protein and plays important roles in multidrug resistance. The focus on serum P-gp levels in this study is a little bit weird.
Thanks for your comment. Indeed, there is sufficient evidence on the function and role of P-gp in multidrug resistance. However, the role of the soluble form in treatment failure and its function in the pathology of some diseases is not known. Therefore, our interest is to provide knowledge on how this soluble form could contribute to the failure of resistance to glucocorticoids in patients with SLE. In addition, it is in our interest to identify biomarkers that are technically easy to measure.
Some studies have validated that the soluble form of P-gp is directly related to membrane expression (Int J Hematol. 2010 Sep;92(2):326-33; Am J Rhinol Allergy. 2016 Jul;30(4):246-9; Biochem Biophys Res Commun.1994 Aug 30;203(1):506-12; Int J Mol Med.2014 Feb;33(2):431-40).
To improve the manuscript, the following lines were added to the discussion section (see lines 250 to 254):
“Some studies have shown a direct relationship between soluble P-gp and its expression at the membrane level [33–36]. The P-gp soluble form has the same molecular weight (170 kDa) as P-gp expressed in the membrane; therefore, it has been proposed that P-gp is released into the extracellular space by a mechanism that does not involve proteolytic cuts [36].”
- It has been shown that P-gp is upregulated on lymphocytes from patients with active SLE and steroids suppress disease activity by inducing apoptosis. The application of high dose steroids might explain the elevation of P-gp in the serum.
We agree with your comment.
GCS can induce P-gp membrane expression through Y-box-protein 1 activation (Histol Histopathol. 2007 Apr;22(4):465-8). However, the increase in the expression of P-gp in turn increases the efflux of GCS from these cells (Arthritis Rheum. 2005 Jun;52(6):1676-83). Besides, inflammation has been described as a possible trigger to increase P-gp membrane expression (Blood. 2002 Sep 1;100(5):1910-2, Immunological Medicine, 44:3, 142-151, Mol. Pharmaceutics 2022, 19, 7, 2327–2334). Therefore, we hypothesize that SR is related to increased P-gp membrane inducted through exposure to GCS and a pro-inflammatory environment.
We added in the discussion section the following text in the lines 268-274:
“Another remarkable result in this study was the positive correlation between P-gp serum levels and GCS doses. Patients with GCS doses ≥20mg/day showed higher P-gp serum levels. These results are consistent with another study [12]. This positive association could be due to the fact that GCS are P-gp membrane inductors and, therefore, can develop an overexpression in the lymphocyte membrane [12]. This phenomenon added to a pro-inflammatory environment can lead to the efflux of different drugs, such as GCS and immunosuppressors from the lymphocytes [39,40].”

Reviewer 2 Report
The authors of the manuscript presented an extensive study on steroid resistance associated with high levels of MIF and P-gp in the serum of patients suffering from SLE. Its a well-described and well-planned work, but I have some questions:
1. Can stress and its elevated level contribute to the occurrence of SLE? on the other hand, can stress translate into the effects of SLE treatment?
2. Why mainly women were included in the study? Is this disease gender-dependent?
3. Whether age, medication and smoking can affect the expression of MIF and P-gp?
4. if and if so how do steroids affect P-gp expression and activity. is there a molecular relationship between P-gp and steroid levels?
5. In section 2.2 (page 3, lines 79-100) the authors showed the figures. I wonder if such large dispersions between the results are correct. Are the results statistically significant with such large differences in the groups? I recommend creating a table or describing the statistical values under figure 1.
6. In figure 1A there is no distribution in the reference group.
7. In line 79, would be Figure 1, but not 1A.
8. In paragraph 2.2 there is no reference to Figures 1A and 1B in the text.
Once these edits and answers have been done, this paper will be ready for publication.
Author Response
Guadalajara, Mexico, October 6th, 2022
We thank the reviewer 2 questions and suggestions regarding to our manuscript molecules-1943860, entitled: Steroid resistance associated with high MIF and P-gp serum levels in SLE patients.
We are submitting the new version of the manuscript above-mentioned. All changes were marked with track changes function in MS Word. All the recommendations and suggestions made by the reviewers were taken into account, and the questions were answered. In the following lines, we describe point by point the details of the revisions to the manuscript and our responses to the referees´ comments.
General modifications:
We submitted the manuscript to the MDPI english editing service, ID: english-51661. This is a new and improved version of the manuscript, with all the corrections made by your MDPI English editing services.
Reviewer 2 questions, observations, and suggestions:
- Can stress and its elevated level contribute to the occurrence of SLE? on the other hand, can stress translate into the effects of SLE treatment?
Thank you for your question.
Evidence shows that trauma and posttraumatic stress disorder are related to a higher incidence of SLE (Arthritis Rheumatol. 2017 Nov;69(11):2162-2169.). However, we did not include stress as a variable. Therefore, we cannot obtain information regarding stress in this study.
- Why mainly women were included in the study? Is this disease gender-dependent?
Thank you for your question. SLE prevalence is higher in women than men, for this reason, we observed an increased number of women compared with men. Furthermore, studies suggests that higher prevalence of SLE in female can be due to endocrine factors (Nat Rev Dis Primers. 2016 Jun 16;2:16039.)
- Whether age, medication and smoking can affect the expression of MIF and P-gp?
Thank you again for your question.
Regarding to P-gp, some studies have demonstrated there is a difference P-gp expression between age and smoking. However, we did not observe age differences between the SR and SS groups, therefore, this variable seems to not be a bias to P-gp serum levels in SR. (Cytometry A. 2011 Nov;79(11):912-9.)
In this study we found a higher prevalence of smoking in the steroid resistance group, furthermore, this association was kept after adjusting by confounding variables, however, when we used the CHAID decision tree, smoking did not show relationship with the steroid resistance. That is why we think the effect of smoking has a relationship with P-gp and MIF, but not with steroid resistance. Furthermore, we performed an analysis not included in the paper where we identified that there is not statistically significance between the patients with tobacco consumption and those without the consumption p= 0.74.
In the discussion section we have added the following text in the lines 259-265:
“Some studies have reported that smoking can modify the expression of P-gp, but there is no evidence of this process with MIF [37,38]. In this study, we identified a major prevalence in smoking in the SR group and this association remained after adjusting the model with confounding variables; however, when we analyzed the CHAID decision tree, this association did not have statistical significance. That is why we think that smoking has a relationship with P-gp and MIF, but not with steroid resistance.”
Regarding medication, there is a large number of drugs that can affect p-gp expression. However, we did not include any drugs that affect p-gp induction, expression and functionality, except for GCS and cyclophosphamide for SLE treatment. However, we did not consider it a confounding variable because we did not find statistical significance between the patients undergoing administration of these medications compared to patients not taking them. In the case of MIF, the medications used by the SLE patients, except for GCS, do not affect their expression (Pharmacol Ther. 2015 May;149:1-123; Immunopharmacol Immunotoxicol. 2015 Apr;37(2):207-13).
- If and if so, how do steroids affect P-gp expression and activity? Is there a molecular relationship between P-gp and steroid levels?
Thank you for your question.
This point has not been elucidated completely, but the evidence shows that steroids, such as other drugs and xenobiotics can stimuli the induce of the P-gp expression in different cells, such as lymphocytes, leading to its efflux and resulting in failure and a subsequent increase of dose of corticoids.
However, the increase in the expression of P-gp in turn increases the efflux of GCS from these cells (Arthritis Rheum. 2005 Jun;52(6):1676-83). Besides, inflammation has been described as a possible trigger to increase P-gp membrane expression (Blood. 2002 Sep 1;100(5):1910-2, Immunological Medicine, 44:3, 142-151, Mol. Pharmaceutics 2022, 19, 7, 2327–2334). Therefore, we hypothesize that SR is related to increased P-gp membrane inducted through exposure to GCS and a pro-inflammatory environment.
We added in the discussion section the following text in the lines 268-274:
“Another remarkable result in this study was the positive correlation between P-gp serum levels and GCS doses. Patients with GCS doses ≥20mg/day showed higher P-gp serum levels. These results are consistent with another study [12]. This positive association could be due to the fact that GCS are P-gp membrane inductors and, therefore, can develop an overexpression in the lymphocyte membrane [12]. This phenomenon added to a pro-inflammatory environment can lead to the efflux of different drugs, such as GCS and immunosuppressors from the lymphocytes [39,40].”
- In section 2.2 (page 3, lines 79-100) the authors showed the figures. I wonder if such large dispersions between the results are correct. Are the results statistically significant with such large differences in the groups? I recommend creating a table or describing the statistical values under figure 1.
Thank you for your interesting question.
Effectively, as you mention in your question, our sample has high variability, which can lead to diminished statistical power; however, the selected and performed tests showed statistical differences (Myors, B., Murphy, K.R., & Wolach, A. (2014). Statistical Power Analysis: A Simple and General Model for Traditional and Modern Hypothesis Tests, Fourth Edition (4th ed.). Routledge). That is why we consider that all the performed statistical tests, including the CHAID decision tree, adequately demonstrate differences, therefore we consider the used test in this study are correct.
Furthermore, according to your recommendation, we added the significance code for the p-value in the footer of the Figure 1, lines 106-107, being the following:
“ *** : p-value ≤0.001; ** : p-value ≤0.01; * : p-value ≤0.05]; ns= no significant.”
- In figure 1B there is no distribution in the reference group.
Due to the characteristics of all the data contained in figure 1B, the distribution in reference group cannot be appreciated in the graphic due to the dimensions of the graphic itself. The range of P-gp serum levels in the reference group is low, that is why it cannot be appreciated in the figure 1B but its described in the line 91 as follows: “range 3.53-14.98 ng/mL.”
- In line 79, would be Figure 1, but not 1A.
Thank you for you observation. We corrected the writing to: “Figure 1 compares MIF and P-gp serum levels between the reference, steroid”, in the above-mentioned line, and added “as well as MIF and P-gp serum levels with different GCS doses” in the lines 80 and 81.
- In paragraph 2.2 there is no reference to Figures 1A and 1B in the text
Thank you for your comment. In the end of line 86 we added the reference for “(Figure 1A)”, and in the line 92 the added text was: “These comparisons are shown in figure 1B.”
Round 2
Reviewer 1 Report
My main concerns have been addressed.